# Research Progress in Transfer, Accumulation and Effects of Microplastics in the Oceans

**Michele Arienzo** [1,*] **, Luciano Ferrara** [2] **and Marco Trifuoggi** [2]

1. Department of Earth Sciences, Environment and Resources, University of Naples Federico II, Via Cintia 21, 80126 Naples, Italy
2. Department of Chemical Sciences, University of Naples Federico II, Via Cintia 21, 80126 Naples, Italy; luciano.ferrara@unina.it (L.F.); marco.trifuoggi@unina.it (M.T.)
* Correspondence: michele.arienzo@unina.it; Tel.: +39-081-2538166

**Abstract:** One of the major concerns regarding the presence of plastics in ocean environments are the effects on marine biota. Plastics can be distinguished in macro- ($\geq$25 mm), meso- (<25 mm–5 mm) micro- (<5 mm–1 μm), and nano-plastic (<1 μm) and are practically omnipresent in aquatic habitats and subject to long-range transport. The purpose of this review is to report the last findings on the release, transfer, accumulation, and effects of micro-plastics, MPs, in the oceans. MPs have the chance to adsorb different kind of organisms and compounds on their outer surface, including bacteria, viruses, algae, and abiotic substances. In this way, they can cause sever hazard once they enter the food chain. Their harm to higher organisms is discussed as well as main routes of MPs–organism interactions, i.e., ventilation, and ingestion. Potential effects on populations, communities, and ecosystems and uptake routes and transition into tissues are discussed. In consideration of the potential threats of plastic particles to ecological functions and human health risks, we recommend specific directions of future research approaches.

**Keywords:** microplastics; marine environment; transfer; accumulation; biological effects

## 1. Introduction

Plastics have become popular for their versatility and convenient usability. The physical-chemical properties of plastics, light weight and durability combined with low production costs, make this material almost irreplaceable in the production of household goods, construction, and industry [1]. Plastics have enormous social benefits because they are very cheap and commonly used by most people for different daily applications. One of the major large applications regards their use in the packaging industry: most consumer products are packaged with plastic wrappers that increase not only their shelf life but also the attraction of the final consumer. One of the major drawbacks of the recent Coronavirus pandemic is represented by the massive usage of single-use plastic, which in normal times amounts to more than 40% of all plastic trash [2]. In 2019, many people were quite confident of the reduced consumption of this material. Most companies, especially those working in the packaging sectors, were converting to green-economy policies to be attractive for younger consumers. The numbers of the disposable plastic consumption, in times of the Coronavirus, are clearly increasing due to the increase in delivery and the use of gloves and masks. The green road that had been taken seems to be slowing down. The multinationals have also been forced to take a step backwards in their environmental protection policies. For example, Starbucks has suspended the use of personal and reusable cups and glasses in Australian and US stores, in favour of disposable containers. A gesture dictated by the desire to make access to points of sale safer, but which weighs incontrovertibly on the amount of waste produced.

### 1.1. What Is the Problem?

#### 1.1.1. Increasing and Significant Presence of Plastic Wastes

World plastics production has significantly increased over the last years and in 2014 it was reported to be 311 million of tons with an estimate of marked increase soon [3]. Fragmented plastics are usually distinguished in macro- (≥25 mm), meso- (<25 mm–5 mm), micro- (<5 mm–1 μm), and nano-plastics (<1 μm) [4]. MPs are omnipresent in aquatic habitats, due to their long distance transport [5,6]. A study reveals that remote areas of the Arctic accumulate large amounts of MP in ice [7]. Plastic pollution is a problem regarding all ocean basins irrespective of developed or underdeveloped regions in the world, Figure 1. Fragments of MPs have been found even in a small marine crustacean in the Arctic Sea [8]. Due to their low density, MPs are transported to coastal regions by waves, tides, and sea currents, eventually accumulating along the coast. The first reports of plastics in the surface ocean were reported at the beginning of the 1970s [9,10]. Other reports describe the presence of plastic fragments in birds in the 1960s [11]. Thus, plastics have been reported to be one of the major primary pollution components of sea environments for a long time; however, the risks associated with their presence have only been recognized and understood later [12]. But plastics can also be biologically decomposed and defragmented by worms and moths as well microbial streams. The streams include *Bacillus* sp. BCBT21, *Bacillus amyloliquefaciens* BSM-1, *B. amyloliquefaciens* BSM-2, *Pseudomonas putida*, *Bacillus subtilis*, *Bacillus cereus*, *Brevibaccillus borstelensis*, *Bacillus vallismortis* bt-dsce 01, *P. protegens* bt-dsce 02, *Stenotrophomonas* sp. bt-dsce03, and *Paenibacillus* sp.bt-dsce04) [13]. If the present trend in production and use continues at this rate, it is quite possible that MPs will overcome fish presence in 2050 [14].

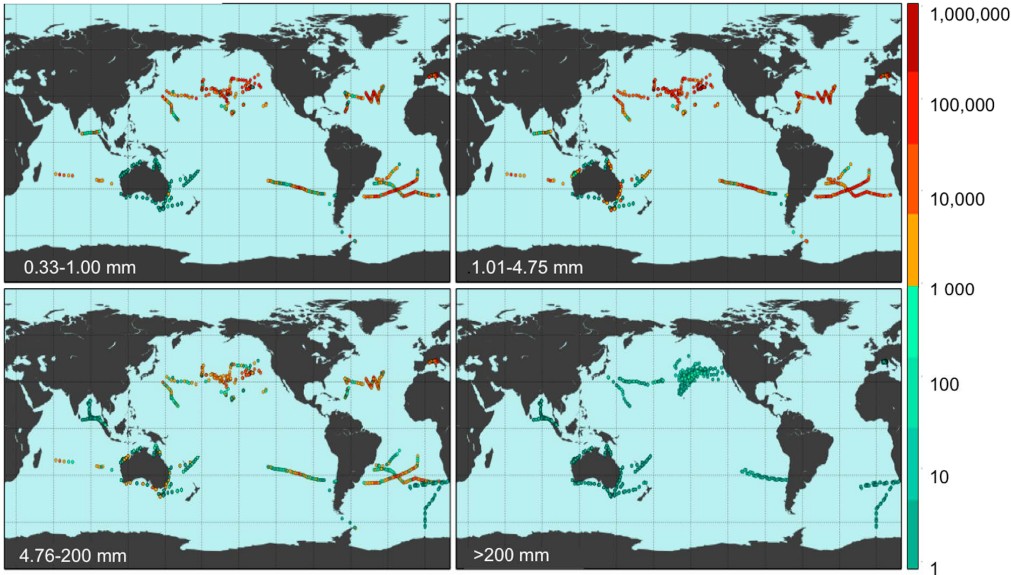

**Figure 1.** Count density (pieces km$^{-2}$; see colorbar) of marine plastic debris measured at 1571 stations from 680 net tows and 891 visual survey transects for each of four plastic size classes (0.33–1.00 mm, 1.01–4.75 mm, 4.76–200 mm, and >200 mm).

#### 1.1.2. Nature of Plastic Wastes

Plastics are made up of macromolecules called 'polymers', which are in turn made up of chains of smaller molecules, called 'monomers'. The different types of plastic differ in terms of their external appearance and intended use, but they have some very specific characteristics in common: they are light, washable, inexpensive, easily malleable once heated, mass-producible, and particularly useful for food preservation. The most common plastics on the consumer products market are polyethylene, PE, used to produce bags, boxes, adhesive tapes, bottles, sweeping bags, pipes, toys, etc., and polypropylene, PP, used for furniture, food containers, detergent bottles, and plastic disposal products [1]. PE



or polyvinyl chloride (PVC) containers abandoned in the environment take between 100 and 1000 years to degrade, while apparently more insubstantial items, such as bags, take at least 1000 years.

New products, heat, and electricity can be obtained by recovery or recycling of plastics. Mechanical recycling involves the transformation from material to material: plastic that is no longer used becomes the starting point for new products. This technique essentially consists of thermal or mechanical reprocessing of plastic waste. Chemical recycling involves returning to the basic raw material by transforming used plastics into monomers of the same quality as the virgin ones, to be used again in production. In practice, the polymers of the different plastics are broken down into their respective monomers through "reverse manufacturing". Non-collected or non-recycled plastics can be destined for energy recovery through the waste-to-energy process. In fact, after a specific selection and shredding treatment, it is possible to obtain alternative fuels used in industrial processes (for example in cement factories) and to produce of thermoelectric energy. Energy recovery involves reusing the energy contained in plastic waste, which has a calorific value comparable to that of coal [1].

The raw materials of plastics are fossil fuels but, in the oceans, naturally occurring biopolymers may be present. The main difference between MPs and biopolymers is that the latter has been always present in the oceans. The fraction of biodegradable plastics significantly increased over the last years. In 2019, it represented about 55% of the overall production of bioplastics [15]. In fact, not all the produced bioplastic is biodegradable since it is commonly mixed with irreplaceable compounds to give commercially functional properties. This means that the destiny of these bioplastics in the environment is uncertain. Bioplastic manufacturing processes need to be improved to provide a clearer understanding of the environmental fate [16]. One advantage of these mixing is due to their lower hydrophobic feature if compared to synthetic plastics.

### 1.1.3. A proper Definition of MPs

One of the issues of the current emerging research on MPs are the lack of a common and universal definition of what MPs are. Most of the international literature define MPs as polymer particles < 5000 μm diameter. This definition makes that MPs represent 92% of plastics in the sea [17], although Eriksen used a smaller breakpoint (4750 μm) [17]. Collectively, MPs vary from 0.1 to 5000 μm, but current knowledge of the dimensional discrimination in the large consumer products is very limited. MPs < 5 mm can form by physical decomposition of larger plastic products or by the emission of so-called microbeads in the environment, which are widely used in the cosmetic industry [18]. The 1972 United States (U.S.) patent for microbeads set the most desirable size range for skin cleaners to 74–177 μm [19].

### 1.1.4. MPs and Sources

MPs are found in the environment in different forms such as pellets, i.e., spherical beads, films, foams, fragments, fibres, etc., as shown in Figure 2. MP fibres are the most frequently reported in literature [3].

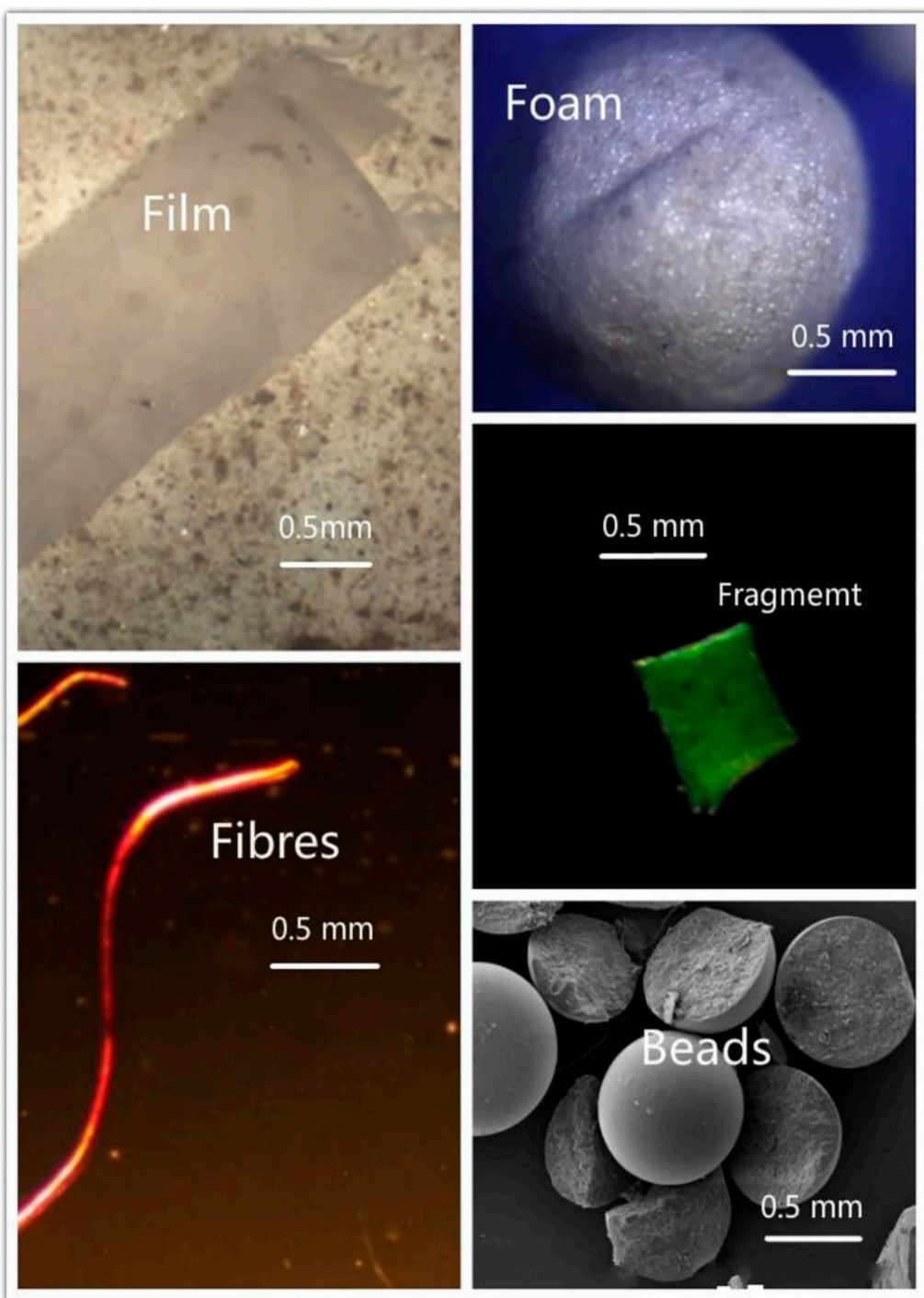

**Figure 2.** Morphologies of MPs in marine environment [20]. With the permission of Elsevier.

MPs are normally of two different types, primary and secondary [21,22]. These two variants were distinguished by the International Research Workshop [23] based on their origin, appearance, effect, fate, and degradation. Primary means that they were originally fabricated in the size of MPs while secondary means that they come from the physical decomposition of larger pieces. Primary MPs are produced during the production or recycling of plastics, blasting and microspheres contained in hygiene and personal care products. Secondary MPs consist of plastic fragments from marine waste, fishing gear, municipal and industrial waste channels, and packaging materials. Primary MPs are produced for domestic or industrial applications that make particular use of their abrasive properties: micro-plastics in cosmetics, toothpastes, deodorants, shaving creams, polishing

pastes, and so on [22,24]. The cosmetics and personal care industry produce intentionally exfoliants from PE, polyethylene terephthalate, PET, polyamides, PA, such as nylon, PP and polymethyl methacrylate, PMMA, and these MPs are in the form of solid particles smaller than 1 mm [18]. The result is that these plastic microspheres are diffused on beaches and in ocean waters around the world [25]. A major concern with PMMA is that once introduced into the environment, it is very difficult to remove them, and they are likely to persist for centuries [26]. Recently, the use of microbeads has been limited in consumer products, like in the U.S., Canada, and the UK. In the U.S., legislation greatly limited the manufacture of microbeads [27]. Virgin resin pellets, with a 5 mm diameter, represent another important source of primary plastics used during the fabrication process of plastic and their subsequent transport.

As we said, secondary MPs originate from the degradation of larger plastic waste that is dumped in the sea (or on land) when exposed to the elements (air, water, light) [22]. This is for instance the case when using modern fishing products, which are among the main sources responsible for pollution from MPs. In this sector, there is a great use of plastic because of the many advantages that this material offers compared to the natural materials traditionally used. The accidental loss, intentional abandonment, or normal wear and tear of fishing gear (ropes, lines, floats, nets, etc.), auxiliary items (fish boxes, gloves, strapping, etc.), and kitchen waste in the marine environment leaves many fibres and plastic fragments [5]. Coastal tourism and recreational activities make extensive use of disposable plastic, thus, contributing heavily to the problem of dispersion of MPs in coastal marine environments [28]. MPs are also released from synthetic garments because of machine washing. It is estimated that a single polyester fibre shirt releases about 1900 fibres in a single wash [29,30]. In recent research, it has been observed that wastewater treatment plants (WWTP) reduce the abundance of MP in effluents by 98% [31]. The specific density of plastic is close to that of water. For this reason, synthetic waste is easily transported through the catchment area into lakes and rivers to finally enter the seas and oceans. Nevertheless, about $8 \times 10^{12}$ MP particles enter the aquatic environment every day through a single wastewater treatment plant [32].

Plastic materials can reach the oceans by different routes, as shown in Figure 3, as riverine input, wastewater effluent, sewage sludge disposal, litter from coastal activities, litter from marine activities, and atmospheric deposition.

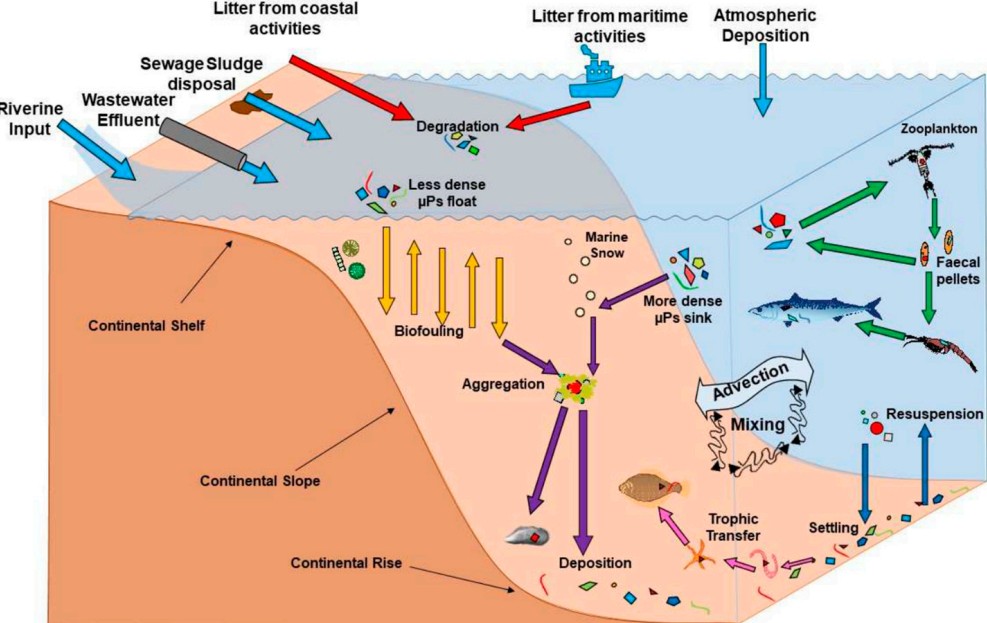

**Figure 3.** MPs pathways in the environment (adapted from [33]).

## 2. MPs Generation

Once in the ocean, plastics can go through fragmentation by UV and mechanical forces and those less dense can float and then go through biofouling, aggregation, and deposit on the marine bed, whereas those denser can sink and directly deposit onto sediments, as shown in Figure 3. Here, MPs can re-suspend and together with suspended MPs can be adsorbed by zooplankton and expulsed as faecal pellets and through the processes of mixing and advection and fish ingestion be included in trophic chains. Environmental degradation processes, such as UV radiation, oxidation, and biodegradation, take hundreds of years to remove MPs from the sea [34]. With extensive weathering, plastics generally develop surface micro-fissures [35] and fragment into progressively smaller particles [36,37]. The dominant cause of degradation of plastics outdoors is solar UV radiation, which facilitates oxidative degradation of polymers [38]. Photodegradation of common plastics such as PE, PP, and polystyrene, PS, are free-radical mediated oxidation reactions [39]. The basic mechanism of this autocatalytic oxidation of common plastics is well established [40]. During advanced stages of degradation, the plastic debris typically discolours, develops surface features, becoming weak and brittle (embrittle) in consequence over time. Any mechanical force (e.g., wind, wave, animal bite, and human activity) can break the highly degraded, embrittled plastics into fragments. Plastic products can incorporate a range of additives selected to modify the properties of the resin to meet the intended product application. These additives change the rate of oxidative degradation (and, therefore, weathering rates) of plastics. For example, UV and heat stabilizers and antioxidants used as additives often markedly retard light-induced degradation of the plastic material. While the weathering of common plastics and their various formulations have been extensively studied in different environments, these studies have historically focused on the early stages of degradation that impact the useful lifetime of the product. Limited information is available on extensive oxidation and fragmentation of highly weathered plastics in the environment. Furthermore, there is virtually no information on weathering of plastics stranded on shorelines, floating in seawater [41] and especially submerged in seawater or sediment. The effects of variables such as mechanical impact, salinity, temperature, hydrostatic pressure, presence of pollutants such as oil in seawater and biofouling (reducing UV exposure) on the rates of weathering according to various types of plastic items are virtually unknown.

MPs of very small size of about 100 microns can show localized micro-fissures on the surface because of UV radiation transmitting into the bulk of the plastic. These micro-fissures represent the preferential route of entrance of MPs into the marine environment. The perpetuation of the oxidation process can give rise to particles at the nanoscale level [42], whose presence has not been fully documented yet. The processes of weathering and fragmentation can be quite relevant and fast on beaches and of several orders of magnitude lower in plastics floating or in the marine sediments. The quantification of plastic degradation in different marine environments remains an open unexplored research area. The only known effect regards the role of very low water temperatures, which hinder weathering of the floating MPs. Another important role is known to be played by the surface fauna, which cause biofouling of floating plastics almost everywhere. Biofouling has the main effect to attenuate the action of solar UV and makes the plastic denser, impeding the UV degradation process [42,43]. There is one side of the ocean where this does not occur or not too much is known and is represented by the aphotic sediment environment. Thus, the production of different MPs sized materials is affected by several environmental factors as well by the specific properties of the polymer. Literature needs to fill the existing gap of how MPs' weathering and fragmentation occur in the oceans, where photo-oxidation, fragmentation, mechanical abrasion, and additive play an important role on MPs' generation.

## 3. MPs Transport

Recent reports reveal that rivers move 1.15–2.41 million tonnes of plastic waste to seas [44] and this is expected to increase in the coming decades. Quantifying the transport of plastic debris from river to sea is fundamental for assessing the risks of plastic debris to the environment. Some authors [44] presented a model approach to evaluate the composition and quantity of MPs flows from European rivers to the oceans. The study evidenced wide differences among European rivers due to socio-economic condition and technical performances of sewage plants. The most sensitive seas to MPs pollution were reported to be the Mediterranean and the Black Sea because of the low MPs removal efficiency of sewage plants. Once generated, MPs can make long trips in the oceans due to several factors like MPs density, airflow, waterpower, etc. MPs debris can accumulate in large-scale subtropical ocean gyres for the effect of convergent ocean surface concentrating currents [45–48]. A similar phenomenon can also occur in a closed sea like the Mediterranean [49], where MPs can remain on the surface because of limited possibility of water exchange. The role of the sea currents varies with depth and, hence, the rates of MPs dispersal are influenced by varying the different circulation flows at different depths. A study by McDonnell and Buesseler [50] reported different sinking rates of plastic debris between 10 and 150 m per day. This means that in the very deep ocean, particles may take up to a year to sediment. Horizontal current might be quite different depending on the water depth, from one metre per second at the surface to a few centimetres a second at 1000 m depth. Thus, a sinking particle may move from its source and along the horizontal pathway from one kilometre (fast sinking rate) to 35 km (slow sinking rate). Depending on marine currents and plastic density, a single plastic particle can continuously sink at the water surface, extending in this way its horizontal displacement. The role of biofouling on density changes in MPs is poorly known. However, sinking and, hence, horizontal displacement can be affected by colonization by macrobiotic fouling organisms [51] of larger items of MPs [52].

## 4. MPs as Vectors and Sink of Biotic and Abiotic Substances

As we have already stated, MPs' outer surface can host bacteria and viruses in the form of biofilms as well as algae and xenobiotics [53]. MPs, together with microorganisms, can be potentially included into the food chain and, hence, be harmful to different groups of organisms [54]. Microbial adhesion and growth might occur due to the specific nature of MPs in terms of composition and characteristics of their outer surface. In fact, MPs might adsorb a wide number of nutrients and organic substances, making them a perfect site for microbial biofilms formation [55]. Some studies report how the formation of biofilm occurs rather quickly on their surface just seven days after their release [56]. MPs with the attached biofilms can also be the pole of attraction of other planktonic organisms becoming a very complex superficial system [57]. This composition can be affected by the water quality, the geographic location, and seasonal variation. It has been estimated that the quantity of adsorbed microorganisms is 1000–15,000 t [58] and this can be verified by advanced microscopical devices like SEM and high-throughput sequencing technology. One of the major risks deriving from this microbial adhesion to MPs is the microbe transportation for long routes through different biogeographic regions and their consequent biological invasion.

MPs can also vehicle xenobiotics on their surface, including persistent inorganic and organic pollutants, like heavy metals, polycyclic aromatic hydrocarbons, and polychlorinated biphenyls. This depends on the properties of the polymers, the surface structure of the weathered plastics, and its specific surface area [55]. Some polymers like PP and PE have more sorption affinity for polycyclic aromatic hydrocarbons and polychlorinated biphenyls than polyethylene terephthalate and PVC. This pollutant sorption process increases the polluting effect of MPs, enabling adsorbed pollutants to travel a long path and reach different and very remote geographical areas [59]. In fact, apart from the chemical additives that are already present at the time of their synthesis, such as phthalates, bisphenol A, and polybrominated diphenyl ethers, MPs can adsorb pollutants. Organic contaminants, metals, and pathogens are taken from the environment and transmitted to the organisms

that ingest them [60]. This aggravates their toxicological profile since with this additional toxic load they can induce greater toxic effects [59]. Moreover, the very small size of these materials means that the total exposed surface area per unit volume is very high, promoting their ability to aggregate and transport toxic substances [61]. The heterogeneous and complex nature of MPs means that various organic and inorganic contaminants can bind to form an "ecocorona", increasing the density and surface charge of the particles and changing their bioavailability and toxicity [62]. In aquatic environments, the surface of nano-plastics and MPs is rapidly coated with various components of natural organic matter (NOM), such as humic substances, extracellular polymeric substances, and proteins [62]. Therefore, the formed eco-corona modulates their bio-reactivity and potential impacts. The aging of the plastic material favours the formation of carbonyl functional groups on the surface [63] and, consequently, the adsorption of organic pollutants is not limited to hydrophobic organic pollutants, but also concerns hydrophilic ones, which have much higher concentrations and are ubiquitous in the environment. Moreover, the interaction of MPs with surfactants can significantly increase their ability to adsorb hydrophilic pollutants [64], promoting the transfer of pollutants from MPs to organisms [65]. The adsorption and the interaction mechanism between aged MPs and hydrophilic organic pollutants represents a new and interesting area of study. Useful in this regard is the use of infrared spectroscopy (IR) and scanning electron microscopy (SEM), techniques able to detect significant surface oxidations and localized micro-fissures on aged MPs.

## 5. MPs in the Marine Environment

One of the major toxic effects of MPs is the interference of these particles in the form of monomers, added additives, i.e., flame retardant and plasticizers, or sorbed pollutants [66]. In addition, the presence of MPs has been shown to alter the composition of the microbial sediment community and nitrogen cycle processes. MPs may become entrained in benthic [67] and pelagic food webs [32]. MPs are persistent and a potential vector of toxic organic compounds to the marine environment and their negative effect on the environment is not only physical, but also chemical due to their capacity to adsorb and accumulate several types of contaminants. The negative relevance of MPs in the marine environment has been recently recognized by its inclusion as a priority descriptor in the Marine Strategy Framework Directive (MSFD) as Descriptor 10. The MPs reduced size and ubiquitous presence in oceans facilitate their ingestion by different organism of the trophic webs [68].

### 5.1. MPs and the Biological Impact

The biological impact of MPs in oceans is an emerging issue. MPs can meet organisms by different ways. One of the most important routes of interaction of MPs is external exposure with the outer surfaces of the organism, including gills and translocated into the organism. It depends on the concentration and size distribution of the MPs and the specific nature of the organism. However, this route is rather small and limited for active feeding organisms in respect to exposure through ingestion. MPs have various sizes and low densities. As a result, many living organisms perceive them as food. Exposure though ingestion is much more important. Several hundred marine species are affected by MPs and at least 10% of them encounter MPs through ingestion [69]. Ingestion has been reported for deposit and suspension feeders [70], crustaceans [71], fish and marine mammals [72], and seabirds [73]. MPs have been identified in the guts or tissues of fish [74], bivalves [75], zooplankton [76], seabirds [77], turtles [78], and whales [79]. Wild freshwater mussels and benthic invertebrates accumulate MPs mainly from sediments, while MPs in non-benthic fish stomachs are mainly from MPs suspended in water. MPs accumulate in digestive and reproductive systems of different trophic freshwater organisms such as *Alella azteca* [80], *Lumbricus variegates* [81], and *Oryzias latipes* [82]. Because their enzymatic system is unable to decompose the plastic, ingestion of the plastic is harmful to organisms and can cause a fatal outcome. MPs can cause intestinal blockage [83], pseudo-satiation resulting in reduced food intake [84], or absorbs and/or release organic pollutants (polybrominated

diphenyl ethers, phthalates, and bisphenol A), [85] as shown in Figure 4. It is also quite frequent that organisms might mistake MPs for prey and ingest them directly [86].

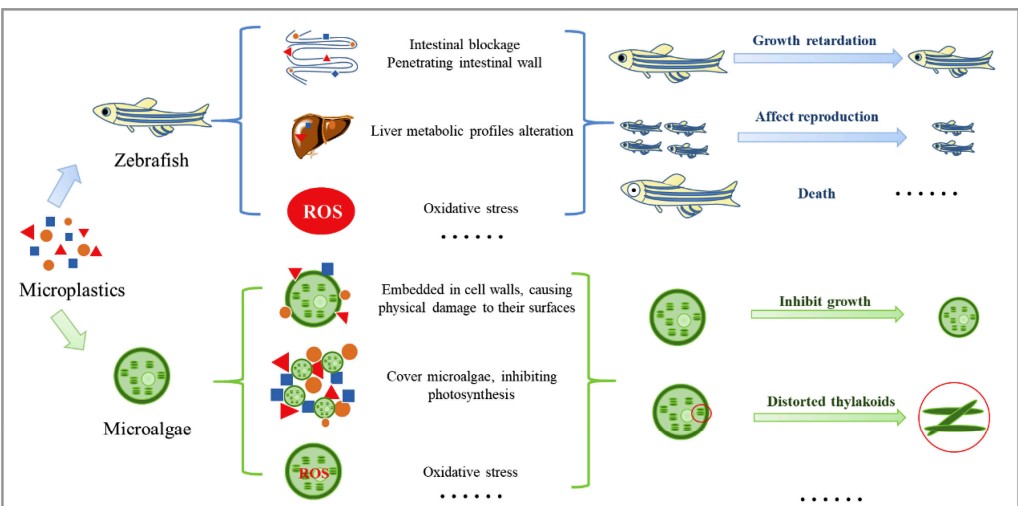

**Figure 4.** Major MPs effects on biota [85] with the permission of Elsevier.

In this way, MPs particles move along the the food web as predators consume prey. MPs in water are frequently mistaken to be food and ingested by plankton feeders, while particles with smaller sizes can also unintentionally be ingested [87]. Thus, ingestion has been described for zooplankton, protozoa, fish, birds, and marine mammals. Field studies have demonstrated that MPs are ingested by a large variety of marine taxa representing various trophic levels, including fish-eating birds, marine mammals, fish, and invertebrates, e.g., lugworms, amphipods and barnacles, mussels, sea cucumbers, and zooplankton. Many invertebrates are able to ingest MPs like deposit feeding lugworms Arenicola marina [34] and sea cucumbers [70], sponges, polychaetes, echinoderms, bryozoans, bivalves, barnacles [88], and detritivores, such as amphipods *Orchestia gammarellus* [34]. Ingestion also regards planktonic organisms like arrow worms and larval fish [10], copepods in laboratory feeding trials [29], invertebrate larvae such as trochophores [29], the echinoderms echinoplutei, ophioplutei, bipinnaria and auricularia [89], and freshwater zooplankton [89]. Most studies have been carried out in a laboratory [90–94]. It is also interesting to note how many marine organisms have a preference for certain MPs' sizes, and this was observed for example by exposure to fluorescent plastic beads. In the case of zooplankton, it has been observed that particle size and abundance regulate the ingestion risk. Moreover, while a large part of MPs are eliminated in an unmodified state by the organism, few particles remain inside the body of the organism. Once MPs particles remain inside, they can produce serious cellular toxic effects [62]. Some studies focused on the presence of particles of polymers in the range of 2–10 mm in several shellfish species and reported that all particles greater than 4 mm remain inside the organism. The uptake of small plastic particles by an organism like zooplankton at the foot of the food chain represents a high risk of transference of these particles to higher organisms. Organisms can directly ingest MPs as well as they can come into contact with MPs once they feed on MPs-loaded prey [95]. However, the risk for organisms of all the food chain level relies not only to the physical damage, but also on the possible release of hazardous xenobiotics vehiculated by MPs due to the release effect of surfactants on the digestive systems [66]. The main characteristic of these compounds is that they are persistent and hydrophobic and, hence, their concentrations increase as they move up a food chain [68]. Their toxic effect varies depending on the nature of the transported chemical. In the case of endocrine-disrupting substances, for example, these can interfere with the endocrine system and on the reproductive capability of organisms [68]. At the moment, the current research does not provide a large number of studies on the accumulation and bio transportability of MPs in living organisms at all

levels including human beings. In the next section, we illustrate the potential effects of MPs in the organisms of oceans.

*5.2. Physical Effects*

Physical adverse effects regard the obstruction or damage of feeding systems [51]. Synthetic fibres may produce intestinal blockage, while hard MPs can damage the intestinal and digestive system. As a consequence, a significant reduction of animal feeding causing even death can be observed [96]. Many authors report cases of physical damages, such as Cole et al., who reported how exposure for a limited time of 24 h with microspheres of PS reduced algae uptake by copepods [97], and Zhang et al., who found that MPs can cause physical damage in algae cell walls [98] and thylakoids compromission in microalgae [99]. MP particles can cause intestinal damage in Danio rerio, with villi and intestinal cell breakage [100]. Microspheres can significantly reduce the fecundity of copepods in the range of concentration of 0.125–25 beads/mL [101]. The main cause of this fecundity reduction can be the reduced food intake. Some studies relate the growth reduction with the reduced fertility [102]. The mechanical damage of MPs for both small and large animals is almost the same once they have been ingested [67]. In the case of zooplankton, MPs block the digestive system [103] and interfere directly or indirectly with their feeding rate [80,104], with the final result of causing an energy deficiency [103]. The physical effect may be related to entanglement, blockage of feeding structures, e.g., salps [105], and zooplankton [97], reduction in the feeding [97], or adsorption of MPs on the organism surface, e.g., algae [106] and zooplankton [97]. Once plastics are ingested and translocated into tissues they can cause toxicity as the case of the sub-lethal or lethal effects observed in certain shearwater species [107,108]. Other authors, like Foekema et al. [109], revealed an unclear relation between the condition factor of some fish species of the North Sea and MPs. In fish, MP accumulation can cause liver glycogen depletion and fat vacuolation [91]. Once ingested, MPs can go through the digestive system and be absorbed by the body, causing inflammation and fibrosis. The xenobiotic adsorbed onto MPs particles can accumulate in tissues and stimulate an immune response [110]. There is a general lack of data in the literature revealing direct particle toxicity effects of MPs translocated from gut to body fluids into organs and cells. Mussels exposed to HDPE plastic powder > 0–80 µm were absorbed by digestive gland vacuoles [111]. In addition to high trophic biota, microalgae, as the primary consumers, can be also affected by the presence of MPs, as shown in Figure 4. The sorption of plastic particles by microalgae results in a reduction in photosynthesis and an increase in reactive oxygen species (ROS). Nano-plastics were found to adsorb onto the surface of *Pseudokirchneriella subcapitata* [112] and hinder the absorption and utilization of photons and $CO_2$ by algal cells [106], but microbeads (10–45 µm PE) were not found to affect plant growth (*Lemna minor*) [113]. Lithner et al. [114] in exposure studies with Daphnia magna found that PVC and polyurethane, PU, produce acute toxicity in Daphnia magna.

*5.3. Chemical Effects*

MPs can cause chemical pollution [92] due to their individual components such as monomers and additives. The constituting monomers are retained biochemically inert due to their high molecular weight [115]. Some studies report how certain monomers have harmful effects. This is the case of the report of Lithner et al., who found that the styrene monomer might have mutagenic or carcinogenic risk, and this is the reason why polystyrene is considered one of the most hazardous polymers and the U.S. Environmental Protection Agency report it as a toxic substance [114]. However, even additives can be very risky, like plasticizers, flame-retardants, antioxidants, and UV stabilizers. Some of them, like PBDEs, are very toxic and are normally added to polymers as ameliorating heat resistance or like nonylphenol as antioxidant [116]. The main reason why these residues can be very harmful for the environment is that they are not tightly bound to polymers and can be released and the release increases with plastic size. The most studied plastic additives are brominated flame-retardants (e.g., polybrominated diphenyl ethers, PBDEs,

used as heat stabilizers, and phthalate plasticizers [114]. Bisphenol A, which is used in the production of polycarbonate, has endocrine disrupting effects that can adversely affect human health [117]. Thus, beside physical damage, the ingestion of MPs may have several ecotoxicological consequences, like the release of chemical additives and/or the transfer and accumulation of organic or inorganic contaminants from seawater to organisms because of ingestion. The release of plastic additives mainly occur where MPs are more concentrated and as an effect of solar irradiation being driven by UV and high temperatures. Plasticizers such as phthalates can affect the reproduction of animals, damage the development of crustaceans and amphipods, and induce genetic aberrations [117]. Their negative effect on reproduction relies on the capability to alter the body's endocrine function [118]. Additives can be also very toxic to plants [119]. Tetrachlorophenol has shown to be directly toxic to phytoplankton [120]. MPs incorporating high molecular weight addives can also significantly alter the sedimentation of diatoms [120]. In the case of humans, MPs can enter the body through different routes, like respiratory tract inhalation, food intake, and skin contact. At the moment, it is currently unsure whether MPs can pass through human skin tissues. Thus, it is more likely that plastics enter the body through the food chain [121]. The main studied effects on human consuming polluted MP fish are reported to be cell necrosis and inflammation, etc. [32]. In vitro studies on cerebral and epithelial human cells reported cytotoxic effects at the cellular level [122]. PBDEs, phthalates, and bisphenol A can interfere with the synthesis of endogenous hormones, and lead to permanent morphological problems in the developmental stage and sexual disorders in adults [123]. Phthalates and bisphenol A could impair the development of crustaceans and amphibians and cause genetic aberrations [124]. In addition to their direct toxicity to organisms, MPs can be vehicles of pollutants from seawater to organisms [125], depending on their size, specific surface area, and hydrophobicity [91]. Wardrop et al. reported that PBDEs could be transferred to fish bodies [126]. The transference of persistent organic pollutants can disrupt critical physiological processes (e.g., cell division, immunity, hormone secretion) and damage organs [67]. Rochman et al. found that fish ingestion of low-density PE containing PAHs, PCBs, and PBDEs caused liver histopathological damages [91]. Rochman et al. [91] reported the transfer of POPs to fish and adverse effects at environmentally concentrations. However, the contribute of adsorbed chemicals to the alteration of body conditions and population-level effects have not been fully investigated. Ingestion of foods containing MPs carrying pollutants can also result in significant accumulation of dangerous compounds along the food chains up to humans. This is the case of persistent organic pollutants accumulation in bird tissue due to consumption of MPs [115]. Accumulation of persistent organic pollutants like polycyclic aromatic hydrocarbons has been observed in Japanese Medaka (*Oryzias latipes*) ingesting MPs with observed toxic effects to the liver as well as metastases and abnormalities [91]. Other studies report how gut surfactants can cause desorption of DDT, phenanthrene and bis-2-ethylhexylphthalate (DEHP)) adsorbed on MPs in marine organisms [127]. The same desorbing effect can also occur in the human intestine. Figure 5 shows the routes of transfer of MPs and their adsorbed pollutants in the food chain. There are studies reporting very fast desorption of pollutants from MPs under simulated intestinal physiological conditions of warm-blooded animals [54] and the same process can also occur in humans, thereby threatening human health.

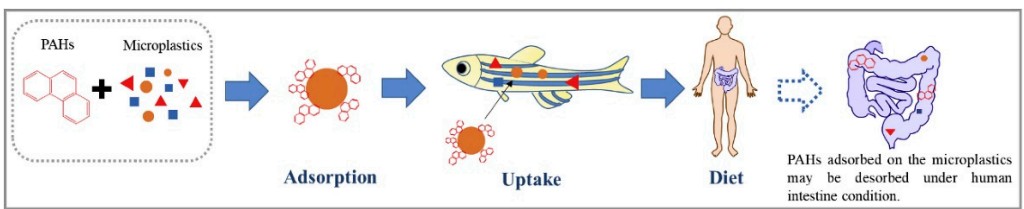

**Figure 5.** Routes o Vehiculation of pollutants in the food chain [85] with the permission of Elsevier.

### 5.4. Populations and Communities Effects

MPs may act as a carrier for the dispersal of biota (invasive species, pathogens) and, hence, in this way may affect marine biodiversity. The effect of exposure of humans and other mammals to MP has been recently reported in the literature [122]. However, the effect of MPs is not limited to the single individual but can have a great potential to alter the structure of the population and the equilibrium of the ecosystem [92]. Negative effects include alteration of the photosynthetic activity and of the consequent productivity of the whole ecosystem. Some studies report how bacteria forming biofilms on plastics are quite different from those present in water [52] and sediment. The presence of biofilm has been detected quite soon after exposure, within one week, and in this way the biofilm can alter the spatial distribution of plastics in the oceans [128] as well these biofilms can constitute food for zooplankton [129]. It is quite interesting to note how plastics wherever it is, water columns or sea-bed sediments, constitute a new habitat and food [52]. Whatever the plastic dimension is, organisms see the plastics as an additional tool to increase their potential to disperse in the oceans [51]. It has been estimated that the colonization of plastic debris in Arctic and Antarctica increased organism dispersion and mobility [51]. This can also represent a growing risk of alien and invasive species diffusion. Plastics can also cause genetic mixing among populations and decreasing genetic variability within populations, as often occurs with seaweeds [130]. The presence of MPs is often detected in mussels and fish caught in coastal waters like the report from Li et al., who reported MPs concentrations in mussel tissue of 0.9–4.6 items per g [131]. Other authors report fish contamination by MPs of 0.2–17.2 items per g [132]. The bioaccumulation of MPs due to their persistence and low biodegradability is reported by Bouwmeester et al. [133].

### 5.5. Uptake and Transition into Tissues

As we have already stated, the literature displays only limited studies on the presence of MPs in the tissues or bloody of organism coming from the field. The majority of the studies regards mussels and polychaetes. Browne et al. [95] found that mussels are capable of ingesting MPs and concentrate the polymer particles in the haemolymph of the circulatory system. In a study with *Mytilus edulis* and *Arenicola marina*, it has been showed there is an average content of $19.9 \pm 4.1$ particles in the lugworm tissue and coelomic fluid, while mussels had an average of $4.5 \pm 0.9$ particles in their tissue [88]. Other studies with mussels [90] have shown the accumulation of PS particles of 3–9 micron in the gut and in the circulatory system within three days of exposure. Von Moos et al. [111] showed the accumulation of HDPE in the size range of >0 to 80 μm in the digestive system and its transition into the lysosomal system. The lysosomal presence cause the collapse of the lysosomal membrane and enzyme emission into the cytoplasm and finally the cell death [111]. Other MPs exposure study [134] reported the transference of polymers from the digestive apparatus to the haemolymph. MPs can enter the cell via different pathways. In the case of eucaryotic cells, particles enter inside the cell by selective diffusion depending on the size of the pores. Then, there are very specialised organelles like coated vesicles, which are destined to the vehiculation of polymers < 200 nm while particles up to 40 micrometres are retained by endocytosis and phagocytosis. Some exposure studies with low dimensional particles reveal nanoparticles can be taken up by the gills and then transferred to the digestive system. Here, particles of nano-sized dimension enter the cells and cause lysosomal damage. Pollutants adsorbed by MPs can be released in certain acidic conditions like those occurring in human intestine after ingestion of contaminated fish, as shown in Figure 5.

### 5.6. Routes of Excretion

The current knowledge about the route of MP excretion is very limited and most studies rely only limited laboratory investigations. Once retained by organisms' haemolymph or tissues, MPs can undergo to two different routes: accumulate or be excreted. This normally depends on several factors, like the dimension of the particle, its morphology

and composition. If the route is accumulation, it is likely that the polymer particle will have a chemical and/or physical effect for an extended period of time. MP concentrations will reach a peak in the haemolymph at a time that will be related to species, plastic type, and exposure time and, subsequently, will follow a decreasing path [95]. In the case the polymer will be excreted, then healing and regeneration process become predominant.

## 6. Future Trends

Most of the current MPs research has focused on trophic transference of MPs alone or in association with xenobiotics to predators under highly contaminated laboratory conditions, thus, quite far from real environment scenarios. This means that future studies need to be carried out under more realistic relevant scenarios, trying to understand the circumstances of occurrence. It remains poorly investigated how consumption of MP contaminated aquatic products influence human health. The main research needs are understating MPs' translocation of non-indigenous species, of weathering-induced fragmentation of PE, PP, and EPS plastics, and the influence of weathering on particle sorption characteristics. There is also the urgency to investigate the fate of nano-sized plastic particles in marine organisms and how they may cross cell membranes and cause cell damage. More research is needed on how the added additives and sorbed xenobiotics may cross the gut wall and assess the risk of harm at an individual and population level. Another important aspect is to consider the species-specific gut conditions that may influence chemical availability and to establish the degree of transfer under different conditions. The transference of MPs along food-webs may create a new and important route for chemical contaminants reaching upper trophic levels, given their significant hydrophobic nature and their consequent capacity to vehicle potential hazardous substances. Innovation on reusable and compostable plastics is the next goal on the road map.

**Author Contributions:** Writing—original draft preparation, review and editing, M.A.; writing—review and editing, L.F. and M.T. All authors have read and agreed to the published version of the manuscript.

**Funding:** This research received no external funding.

**Institutional Review Board Statement:** Not applicable.

**Informed Consent Statement:** Not applicable.

**Conflicts of Interest:** The authors declare no conflict of interest.

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
