# Peer review of "Research Progress in Transfer, Accumulation and Effects of Microplastics in the Oceans"

_jmse, doi:10.3390/jmse9040433_

Round 1

Reviewer 1 Report

Dear authors,

The need of scientific papers regarding microplastic presence and classification in natural environments is still needed and there are still many questions to be asked and, finally, answered. As this research field is under scientific development and under a great lack of details I bring in front following requests, as I see this review with potential:

  1. I recommend a strong improving of the English grammar, punctuation and consistency of the used verb tense (have/has). Entire manuscript has different types of mistakes and it needs a proper proof check.
  2. Few misleading information as follows:

    rows 64 - 65 - there are acknowledge several bacterial species which can degrade different polymers (Bacillus sp. BCBT21, Bacillus amyloliquefaciens BSM-1, B. amyloliquefaciens BSM-2, Pseudomonas putidaBacillus subtilisBacillus cereusBrevibaccillus borstelensisBacillus vallismortis bt-dsce 01, P. protegens bt-dsce 02, Stenotrophomonas sp. bt-dsce03, and Paenibacillus sp.bt-dsce04) from Chandra et al., 2020. Microplastic degradation by bacteria in aquatic ecosystem. Microorganisms for Sustainable Environment and Health. 431-467.

    row 306: Horizon 2020 could not "identify marine litter among the principal objectives of Marine Strategy Framework Directive (2008/56/EC)". Horizon 2020 was a EC research funding program that allow different projects to bring in the sphere of knowledge research opportunities and novelty to science. Probably, 2008/56/EC was a launched call that considered marine litter among the principal objectives of Marine Strategy Framework Directive, and, therefore marine litter is considered one of the biggest concerns of the EC.

  3. Very low resolution of the figures 2 - 6, making the images unreadable. The paper need understandable writing on images/figures.
  4. Using the full names of the polymers is needed only the first time, the second time is necessarily only the abbreviation. Also, PU has not been explained (row 412) - "polyuretan" is missing.
  5. Lack of consistency using the abbreviation MPS of term microplastics. I recommend using only once the full ward, then only the abbreviation (excepting the abstract);
  6. I recommend to rethink the order of the chapters/subchapters (also their titles), as the manuscript is planned in a some-how chaotic form - for example I propose the subchapter 1.1.3 to replace 1.1.1; chapter 4 to be included as a subchapter in ch. 5
  7. Different types of light mistakes as follows:
    1. row 28 - "durability" is synonym with "strength";
    2. r. 43 - needs citation after "United States";
    3. r. 81-82 - needs rephrasing; disposal cannot be done by recovery;
    4. r. 95 - personal hygiene products* - to be added
    5. r. 98-100 - needs rephrasing "The fraction...";
    6. r. 102-105 - Last 3 sentences of this paragraph needs corrections/rephrasing - not understandable;
    7. r. 108 - MPs are;
    8. r. 110 - Eriksen - correct citation needed;
    9. r. 138 - "and so on" doubled;
    10. r. 286 - I don't recommend using "..." as exampled there;
    11. r. 331 - citation needed after "polybrominated diphenyl ethers, phthalates, and bisphenol A"
    12. r. 533 - limited;

Author Response

Reviewer 1:

1)    I recommend a strong improving of the English grammar, punctuation and consistency of the used verb tense (have/has). Entire manuscript has different types of mistakes and it needs a proper proof check.

Answer: the text was improved for English grammar, punctuation, and consistency of the verb tense;

2)    Page 2, rows 64 - 65 - there are acknowledge several bacterial species which can degrade different polymers (Bacillus sp. BCBT21, Bacillus amyloliquefaciens BSM-1, B. amyloliquefaciens BSM-2, Pseudomonas putida, Bacillus subtilis, Bacillus cereus, Brevibaccillus borstelensis, Bacillus vallismortis bt-dsce 01, P. protegens bt-dsce 02, Stenotrophomonas sp. bt-dsce03, and Paenibacillus sp.bt-dsce04) from Chandra et al., 2020. Microplastic degradation by bacteria in aquatic ecosystem. Microorganisms for Sustainable Environment and Health. 431-467.

Answer: The reviewer suggestion was considered very useful, and his whole statement was included inside the manuscript at page 2, raws 64-70 including also the proposed reference;

Page 9 row 306: Horizon 2020 could not "identify marine litter among the principal objectives of Marine Strategy Framework Directive (2008/56/EC)". Horizon 2020 was a EC research funding program that allow different projects to bring in the sphere of knowledge research opportunities and novelty to science. Probably, 2008/56/EC was a launched call that considered marine litter among the principal objectives of Marine Strategy Framework Directive, and therefore marine litter is considered one of the biggest concerns of the EC.

Answer: the wrong statement was replaced with a new phase: The negative relevance of MPs in the marine environment has been recently recognized by its inclusion as a priority descriptor in the Marine Strategy Framework Directive (MSFD) as Descriptor 10.

3)    Very low resolution of the Figures 2 - 6, making the images unreadable. The paper needs understandable writing on images/figures. 

Answer: the overall quality of the Figures was improved though using original images obtained by Elsevier. It was also included a new Figure, Figure 1 of the new list, as the reviewer 2 asked for a map of microplastic distribution in the Oceans. We also deleted Figures 2 and 4 of the old list. 

4)     Using the full names of the polymers is needed only the first time, the second time is necessarily only the abbreviation. Also, PU has not been explained (row 412) - "polyuretan" is missing.

Answer: the abbreviation and the full name of polymers was specified throughout all the text as suggested by the reviewer;

5)    Lack of consistency using the abbreviation MPS of term microplastics. I recommend using only once the full ward, then only the abbreviation (excepting the abstract);

Answer: the term MPs replaced the word microplastics in all the manuscript except when it was first cited.

6)    I recommend rethinking the order of the chapters/subchapters (also their titles), as the manuscript is planned in a some-how chaotic form - for example I propose the subchapter 1.1.3 to replace 1.1.1; chapter 4 to be included as a subchapter in ch. 5.

Answer: we do not agree with this observation since the chapters and subchapters are listed in a logical order: in fact. In 1.1.1. there is a description of the overall problem of plastics (not microplastics) ocean contamination, with data on their nature, diffusion and concern about their constant and continuous release in the environment. In 1.1.2 it is explained what are microplastics, their chemical nature and the major kind of microplastics in the oceans. In 1.1.3. it is explained how microplastics are classified.  

Different types of light mistakes as follows:

row 28 - "durability" is synonym with "strength";
r. 43 - needs citation after "United States";
r. 81-82 - needs rephrasing; disposal cannot be done by recovery;
r. 95 - personal hygiene products* - to be added
r. 98-100 - needs rephrasing "The fraction...";
r. 102-105 - Last 3 sentences of this paragraph needs corrections/rephrasing - not understandable;
r. 108 - MPs are;
r. 110 - Eriksen - correct citation needed;
r. 138 - "and so on" doubled;
r. 286 - I don't recommend using "..." as exampled there;
r. 331 - citation needed after "polybrominated diphenyl ethers, phthalates, and bisphenol A"
r. 533 - limited;
 Answer: All the light mistakes were considered, and the text consequently amended. 

Reviewer 2 Report

The review is very important in the field of environmental toxicology. However, it is necessary to include at least one table indicating the types of microplastics, the most polluted oceans and the origin.

Author Response

Reviewer 2:

The review is very important in the field of environmental toxicology. However, it is necessary to include at least one table indicating the types of microplastics, the most polluted oceans and the origin.

Answer: We do think that it is not necessary to include a Table for the types of MPs since their nature is well illustrated in the manuscript. We included a new Figure, Figure 1, illustrating the overall worldwide problem of MPs pollution in the oceans.

Reviewer 3 Report

The paper „Research progress in transfer, accumulation, and effects of microplastics in the oceans” is the paper review which presents the general and some specific findings of the microplastic particles occurrence in marine environment. Authors presented the problem taking into account the issues related to the current plastic production, waste disposal, release to the environment, transfer, accumulation, and harm effects of microplastics to organisms on different trophic level. They pay attention to the fact that despite introduction a new regulations on plastic wastes, the current pandemic situation forces the return of some single-use products again, which, presumably, will not limit the release of plastic waste to the water environment. In the paper are presented characteristic physical and chemical properties of microplastic particles, the changes they undergo in aquatic environment and their potential toxic impact on different organisms, including human. Authors of the paper determined the new important directions of further investigations to explain the real and complex effects of microplastics on living organisms and activities which should be carried out to prevent water contamination with plastic waste.

The discussed issue of the microplastic presence in the marine environment is very extensive and multidirectional, therefore, the authors presented selected and important issues from their point of view. According to the authors’ suggestions, future studies which should be carried  "under more realistic relevant scenario"  may be difficult to implement, due to the complexity of the factors influencing the ongoing processes.

The content of the paper is described clearly and it is understandable for the reader. The figures: 2, 3, 4, 5, 6 require improvement in quality, they are too small and illegible. The text of the article should be checked by a native speaker.

After correction the paper can be published. 

Author Response

The figures: 2, 3, 4, 5, 6 require improvement in quality, they are too small and illegible. The text of the article should be checked by a native speaker.

Answer: As already stated above, the overall quality of the Figures was improved using better resolution images and enlarging most of the images as well as the text was improved for the English.   

Round 2

Reviewer 1 Report

Dear authors,

the review paper is at an almost publishing shape, few details must be cared for. Please consider below-mentioned comments.

Check the consistency of using "MPs" and "MPS"
I recommend the term micro-fissures instead of cracks or microcracks.

Please use higher resolution (or higher fonts of the text therein) for figures in order to be intelligible for an A4 style, especially for fig 3, 4, 5

row 65-68 there are many studies on different type of plastic defragmentation made by different species excepting bacteria (worms, moths, etc). You can chose mentioning other species as well, or just exemplifying the bacteria.

row 83-84 I recommend adding citation, also few words about MP fragmentation and persistence in larger area as well, as a consequence of the fragmentation.

row 105-108 paragraph needs rephrasing  "This makes that the destiny of these bioplastic in the environment blends is uncertain. Thus, the preparation of these mixing need to be improved for a clearer understanding of their environmental fate [16]. One advantage of these mixing is due to their lower hydrophobic feature if compared to synthetic plastics." 

row 111 - ...what MPs are.

row 165 - effluents

row 226 - most of the MPs derive from land areas through rivers. It is mandatory to mention riverine way of transport in this chapter/subchapter (e.g. Pojar et al., 2021, Siegfried et al., 2017,  Leslie et al., 2017) 

row 348 - observed

row 358 - ...relies not...\

row 361 - that

row 361-362 - "and hence biomagnificate along the food chain" need further explanation and citation

row 371 - needs rephrasing: "The physical adverse effects regard physical obstruction or damage of feeding appendages or digestive tract or other physical harm"

row 375 - ...damages as Cole et al...

row 484 - Arctic and Antarctic islands - please check. Arctic is not an island and Antarctica is a continent.

Author Response

Answer to comments:

  1. Check the consistency of using "MPs" and "MPS"
    I recommend the term micro-fissures instead of cracks or microcracks.

Answer: the consistency of using MPs in the place of MPS was checked and MPs replaced in all the manuscript.

  1. Please use higher resolution (or higher fonts of the text therein) for figures in order to be intelligible for an A4 style, especially for fig 3, 4, 5

Answer: we used in the manuscript the original figures we obtained from Elsevier. They are already in the format and in the legibility as they have been sent us. We cannot improve them.

  1. row 65-68 there are many studies on different type of plastic defragmentation made by different species excepting bacteria (worms, moths, etc). You can chose mentioning other species as well, or just exemplifying the bacteria.

Answer: This sentence was replaced with: Beside this strong accumulation, there are many studies on different type of plastic defragmentation made by worms and moths as well as by microbial streams. The streams include Bacillus sp. BCBT21, Bacillus amyloliquefaciens BSM-1, B. amyloliquefaciens BSM-2, Pseudomonas putida, Bacillus subtilis, Bacillus cereus, Brevibaccillus borstelensis, Bacillus vallismortis bt-dsce 01, P. protegens bt-dsce 02, Stenotrophomonas sp. bt-dsce03, and Paenibacillus sp.bt-dsce04) [13].

  1. row 83-84 I recommend adding citation, also few words about MP fragmentation and persistence in larger area as well, as a consequence of the fragmentation.

Answer: Citation was added, namely [1]. A paragraph on MPs fragmentation and persistence is already present see lines 166-183.

  1. row 105-108 paragraph needs rephrasing  "This makes that the destiny of these bioplastic in the environment blends is uncertain. Thus, the preparation of these mixing need to be improved for a clearer understanding of their environmental fate [16]. One advantage of these mixing is due to their lower hydrophobic feature if compared to synthetic plastics." 

Answer: the paragraph was rewritten as: ‘This means  that the destiny of these bioplastic in the environment  is uncertain. Bioplastic manufacturing processes need to be improved to have clearer understanding of the environmental fate [16]. One advantage of these mixing is due to their lower hydrophobic feature if compared to synthetic plastics.’

  1. row 111 - ...what MPs are.

Answer: the replacement was done.

  1. row 165 – effluents

answer: the replacement was done.

  1. row 226 - most of the MPs derive from land areas through rivers. It is mandatory to mention riverine way of transport in this chapter/subchapter (e.g. Pojar et al., 2021, Siegfried et al., 2017,  Leslie et al., 2017) 

Answer: a new paragraph was added including the reference of Siegfried et al: Recent reports reveal that rivers move 1.15-2.41 million tonnes of plastic waste to seas [44] and this is expected to increase in the coming decades. Quantifying the transport of plastic debris from river to sea is fundamental for assessing the risks of plastic debris to the environment. Some authors [44] presented a model approach to evaluate the composition and quantity of MPs flows from European rivers to the oceans. The study evidenced wide differences among European rivers due to socio-economic condition and technical performances of sewage plants. The most sensitives seas to MPs pollution were reported to be the Mediterranean and Black Sea because of the low MPs removal efficiency of sewage plants.

  1. row 348 – observed

Answer: the amendment was done.

  1. row 358 - ...relies not...\

Answer: the amendment was done.

  1. row 361 – that

Answer: the amendment was done.

  1. row 361-362 - "and hence biomagnificate along the food chain" need further explanation and citation

Answer: the rephrasing was: The main characteristic of these compounds is that they are persistent, hydrophobic and hence their concentrations increase as they move up a food chain [68].

row 371 - needs rephrasing: "The physical adverse effects regard physical obstruction or damage of feeding appendages or digestive tract or other physical harm"

Answer: we rephrased as follows: Physical adverse effects regard obstruction or damage of feeding systems [51].

  1. row 375 - ...damages as Cole et al...

Answer: the amendment was done.

  1. row 484 - Arctic and Antarctic islands - please check. Arctic is not an island and Antarctica is a continent.

Answer: we rephrased as follows:  It has been estimated that the colonization of plastic debris in Arctic and Antarctica increased organism dispersion and mobility [51].